# HATSolver: Learning Gröbner Bases with Hierarchical Attention Transformers

**Mohamed Malhou**
FAIR, Meta Superintelligence Labs
Sorbonne Université, CNRS, LIP6
F-75005 Paris, France
mmalhou@meta.com

**Ludovic Perret**
EPITA, EPITA Research Lab (LRE)
Le Kremlin-Bicêtre, France
ludovic.perret@epita.fr

**Kristin Lauter**
FAIR, Meta Superintelligence Labs
klauter@meta.com

## Abstract

At NeurIPS, Kera et al. (2024) introduced the use of transformers for computing Gröbner bases, a central object in computer algebra with numerous practical applications. In this paper, we improve this approach by applying Hierarchical Attention Transformers (HATs) to solve systems of multivariate polynomial equations via Gröbner bases computation. The HAT architecture incorporates a tree-structured inductive bias that enables the modeling of hierarchical relationships present in the data and thus achieves significant computational savings compared to conventional flat attention models. We generalize to arbitrary depths and include a detailed computational cost analysis. Combined with curriculum learning, our method solves instances that are much larger than those in Kera et al. (2024).

## 1 Introduction

Systems of multivariate non-linear equations are ubiquitous in mathematics and its applications, emerging naturally in fields as diverse as cryptography, coding theory, optimization, computer vision, biology, etc . . . e.g. Perret (2016); Colotti et al. (2024); Fontán et al. (2022); Buchberger (2006); Wang & Xia (2005); Boulier et al. (2011). In contrast to systems of linear equations, solving a system of multivariate non-linear equations, also known as the PoSSo problem, is a well-known, NP-hard Garey & Johnson (1979), computationally hard problem.

Among existing techniques, Gröbner bases Cox et al. (2007); Buchberger (1965; 2006) is the most widely used approach for solving PoSSo. Indeed, the set of common solutions of a polynomial system – also known as its variety – is studied via the ideal generated by those polynomials. Gröbner basis provides a canonical generating set of a polynomial ideal, allowing in particular to efficiently solve PoSSo. More generally, Gröbner basis is a powerful tool enabling to address a wide range of problems related to polynomial ideals Cox et al. (2007), such as membership testing (determining if a polynomial belongs to an ideal), elimination (reducing systems to fewer variables), finding algebraic relations (syzygies) among generators of an ideal, etc . . .

The versatility of Gröbner bases renders their computation inherently difficult but also appealing. From a theoretical point of view, this is illustrated by the folklore result about the double-exponential worst-case complexity for computing a Gröbner basis Mayr & Meyer (1982). This theoretical statement holds for a very peculiar example and does not fully capture the actual hardness of solving PoSSo in practice. In particular, we can usually assume that the variety is radical and has a finite number of solutions. In such a setting, the worst-case complexity drops to single exponential.

From an algorithmic point of view, the historical method for computing Gröbner bases was introduced by Buchberger in his PhD thesis Buchberger (1965; 2006). Over the past 25 years, significant improvements have been made, leading to more efficient algorithms such as $F_4$ and $F_5$

Faugère (1999; 2002) which are now implemented in major computer algebra systems, such as Maple, Magma or Singular, and open-source projects such as `Msolve` Berthomieu et al. (2021).

Although tremendous progress has been made, Gröbner bases remain computationally difficult, and their numerous applications make the design of efficient algorithms both challenging and rewarding. Notably, the security of cryptographic standards such as `AES` Cid et al. (2006); Steiner (2024) or new lattice-based post-quantum standards hinges directly on the hardness of solving algebraic equations Cid et al. (2006); Albrecht et al. (2014); Steiner (2024).

Recent works, e.g. Peifer et al. (2020); Kera et al. (2024; 2025), started to explore advanced machine learning techniques for polynomial system solving. Peifer et al. (2020) employs reinforcement learning to perform $S$-pair selection, a critical step in Buchberger's algorithm. Kera et al. (2025) uses deep learning to identify and eliminate computationally expensive reduction steps during the computation of Border bases; another fundamental tool for solving systems of equations Kehrein & Kreuzer (2005). Most notably, Kera et al. (2024) demonstrates that Transformer models are capable of learning Gröbner bases computation Kera et al. (2024). The authors propose reframing the problem as a supervised learning task, where a model is trained on pairs of polynomial systems and their corresponding Gröbner bases. To enable this, they address two previously unexplored algebraic challenges of efficiently generating random Gröbner bases, and constructing diverse non-Gröbner sets that generate the same ideal as a given Gröbner basis (the "backward Gröbner problem"). Their solution focuses on 0-dimensional radical ideals, which are common in applications.

A notable limitation of the approach presented in Kera et al. (2024) is its restricted scalability to larger polynomial systems. In their experiments, the authors were only able to handle systems with up to five variables ($n \leq 5$). Moreover, they had to significantly reduce the density of the polynomial systems for $n = 3, 4$, and 5—meaning that the input polynomials were made much sparser. This reduction in density ($\rho << 1$) was necessary to avoid overwhelming the model and hardware with excessively long input and output sequences, a direct consequence of the quadratic memory and computational cost of the attention mechanism in standard Transformers. As a result, the method has not been demonstrated on denser or higher-dimensional systems, highlighting a key scalability bottleneck that must be addressed.

## 1.1 Main Results

To overcome this challenge, we propose replacing the multi-head attention layer in the Transformer encoder—the most computationally intensive component of the model—with a hierarchical attention mechanism that leverages the inherent tree-like structure of multivariate polynomial systems. The hierarchical attention layer operates in two distinct stages: bottom-up and top-down. In the bottom-up phase, attention is computed locally at each hierarchical level, beginning at the term level ($\ell = 0$), progressing to the polynomial level, and culminating at the system level. Subsequently, in the top-down phase, information is propagated back to the leaf nodes using various strategies, including cross level attention and simple additive aggregation. This hierarchical approach significantly reduces the sequence lengths processed by the attention layers, resulting in substantial computational cutbacks as the problem dimensions grow. Notably, we successfully computed Gröbner bases for systems of up to 13 variables and degree 11, which compares favorably with efficient tools such as `Msolve` and `STD-FGLM` (see Table 1).

To accelerate the model's learning process, we employ a curriculum learning strategy, training the model on datasets with progressively increasing levels of difficulty and problem sizes. We validate the merit of this method by applying it to the base model of Kera et al. (2024), enabling it to solve systems with $n = 7$ variables at full density. This surpasses the previous results of Kera et al. (2024), where the largest reported success was limited to $n = 5$ variables and a density of only $\rho = 0.2$.

## 2 Preliminaries

### 2.1 Self Attention

Self-attention is a fundamental mechanism in transformer architectures Vaswani et al. (2017), enabling models to dynamically contextualize each element of an input sequence by attending to all other elements. This mechanism is crucial for capturing long-range dependencies and modeling

complex relationships within sequential data, which is essential for tasks in natural language processing, vision, and beyond.

Given a set of queries $Q \in \mathbb{R}^{s \times d}$, keys $K \in \mathbb{R}^{l \times d}$, and values $V \in \mathbb{R}^{l \times d}$, the self-attention mechanism computes a similarity score between each query and all keys:

$$s(Q, K) = \text{softmax}(\frac{QK^\top}{\sqrt{d}}) \in \mathbb{R}^{s \times l}$$

The softmax function normalizes the scores across all keys for each query, converting them into a probability distribution that sums to 1.

The output of the attention layer is a weighted sum of the values $V$, where the weights are the attention scores:

$$\text{Att}(Q, K, V) = s(Q, K)V \in \mathbb{R}^{s \times d}$$

In the context of hierarchical or structured data, such as trees or graphs, self-attention can be applied to sets of embedding vectors $\boldsymbol{E} \in \mathbb{R}^{n \times d}$ corresponding to the leaves or child nodes of a parent node. The embedding vector for the parent node can then be computed by a pooling function $p(\boldsymbol{E})$, which aggregates the information from its children. Common pooling strategies include mean pooling and selection of a specific child embedding:

$$p(\boldsymbol{E}) = e \in \mathbb{R}^d \quad \text{e.g.} \quad p(\boldsymbol{E}) = \frac{1}{n} \sum_{k=1}^{n} \boldsymbol{E}_{k,:} \quad \text{or} \quad p(\boldsymbol{E}) = \boldsymbol{E}_{0,:}$$

We tensorize the attention function $\text{Att}(Q, K, V)$ and pooling functions $p$ by extending them to operate on tensors with an arbitrary number of leading dimensions, which typically represent batch size or other contextual groupings.

## 2.2 GRÖBNER BASES

Let $k$ be a field, $k[\mathbf{x}] = k[x_1, \ldots, x_n]$ be the polynomial ring in $n$ variables over $k$. Let $I$ be the ideal $I = \langle f_1, \ldots, f_m \rangle = \{\sum_{i=1}^{m} h_i f_i \mid h_1, \ldots, h_m \in k[x_1, \ldots, x_n]\} \subseteq k[\mathbf{x}]$ generated by $f_1, \ldots, f_m \in k[\mathbf{x}]$. A finite set $G = \{g_1, \ldots, g_t\} \subset I$ is called a **Gröbner basis** Cox et al. (2007); Buchberger (1965; 2006) for $I$ with respect to an admissible monomial order $\prec$ (see appendix A.1) if:

$$\langle \text{lt}(g_1), \ldots, \text{lt}(g_t) \rangle = \langle \text{lt}(I) \rangle$$

where $\text{lt}(I) = \{\text{lt}(f) \mid f \in I \setminus \{0\}\}$ is the set of leading terms of all nonzero polynomials in $I$ and the notation $\langle S \rangle$ refers to the ideal generated by the set $S$.

As defined below, a Gröbner basis is not unique, which motivates introducing the concept of **reduced** Gröbner basis. $G$ is reduced if each $g_i$ is monic and and no monomial of $g_i$ lies in $\langle \text{lt}(G \setminus \{g_i\}) \rangle$. For any ideal $I$ and monomial order $\prec$, there exists a unique reduced Gröbner basis.

## 2.3 SHAPE POSITION SYSTEMS

Gröbner bases is a fundamental computational tool in computer algebra that provide algorithmic solutions to fundamental problems such as computing the variety associated to $I = \langle f_1, \ldots, f_m \rangle \subseteq k[\mathbf{x}]$. The definition of Gröbner basis, and their properties, depend on the monomial ordering.

In particular, it can be proved that a lexicographic Gröbner basis of a zero-dimensional (i.e. finite number of solutions) radical ideal has a triangular shape, which generically is as follows:

$$G = \{h(x_n), x_1 - g_1(x_n), x_2 - g_2(x_n), \ldots, x_{n-1} - g_{n-1}(x_n)\}, \tag{1}$$

where $h, g_1, \ldots, g_{n-1} \in k[x_n]$ are univariate polynomials in the last variable $x_n$, with $\deg g_i < \deg h$ for all $i = 1, \ldots, n-1$.

$I \subseteq k[\mathbf{x}]$ is said to be in **shape position** if its reduced Gröbner basis has the triangular form as in equation 1. Following Kera et al. (2024), we will restrict our attention to ideals such that their Gröbner basis is in shape position.

## 2.4 DATASET GENERATION

We adopt the backward generation introduced by Kera et al. (2024) to construct training datasets consisting of pairs $(F, G)$ where $F$ is a random looking system of polynomial equations and $G$ is its corresponding reduced Gröebner basis which is in shape position. Kambe et al. (2025) proved that the samples generated by this algorithm are sufficiently general, ensuring a rich and diverse training set. The backward technique proceeds as follows: 1) draw $h, g_1, \ldots, g_{n-1}$ uniformly at random from $k[x_n]_{<d}$ subject to the degree condition above and 2) generate *non-Gröbner* training inputs $F = \boldsymbol{U}_1 \boldsymbol{P} \bar{\boldsymbol{U}}_2 G$ by multiplying $G$ with random unimodular upper-triangular matrices $\boldsymbol{U}_1$ and $\boldsymbol{U}_2$ and a permutation matrix $\boldsymbol{P}$.

The resulting dataset is balanced: each $(F, G)$ pair satisfies $\langle F \rangle = \langle G \rangle$, $F$ is *not* a Gröbner basis, yet $G$ is. This algorithm as described above generates systems with $|F| = |G| = n$ with $n$ the number of variables. However, the actual algorithm of Kera et al. (2024) generates $s \geq n$ equations by using $U_2 \in k[x_1, \ldots, x_n]^{s \times n}$ rectangular unimodular upper-triangular matrix. We refer the reader to Kera et al. (2024) Section 4.3 and Kambe et al. (2025) for more details.

## 3 METHOD

### 3.1 LIMITATIONS OF FLAT ATTENTION

Writing a system of multivariate polynomial equations as a sequence of tokens grows rapidly with the number of variables and the total degrees of the equations. For instance, the number of different monomials with $n = 5$ variables and total degree $d \leq 10$ is $\binom{d+n}{d} = 3003$. A system of $n+2$ equations could therefore contain over $7 \cdot 3003 \cdot 7 = 147147$ tokens, assuming each term is encoded using n+2 tokens (see section 3.4). This explosion in token count makes training a sequence-to-sequence model, such as a transformer, on such data particularly challenging. However, these systems of equations are highly structured and can naturally be represented as trees. This raises the question of whether attention-based models can be adapted to tree-like structures: rather than having each token attend directly to every other token, a token could attend primarily to its siblings (i.e., those sharing the same parent) and to other tokens indirectly through their parent nodes, and so on.

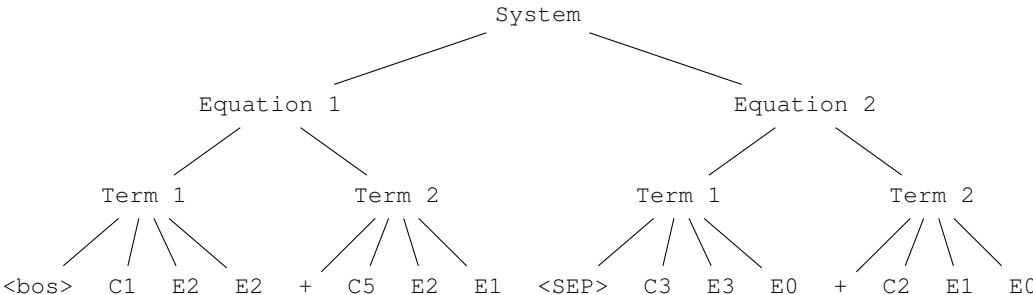

Figure 1: Hierarchical representation of the tokenized system of equations $p_1 = x_0^2 x_1^2 + 5x_0^2 x_1$, $p_2 = 3x_0^3 + 2x_1$ over $k[x_0, x_1]$.

### 3.2 HIERARCHICAL ATTENTION LAYER

The hierarchical attention mechanism operates in two successive phases. The first phase involves the computation of local attention at each hierarchical level, with information being propagated in a bottom-up manner through the hierarchy.

Let us denote $\mathbf{X} = \mathbf{X}^{(0)} \in \mathbb{R}^{\ell_{n-1} \times \cdots \times \ell_1 \times \ell_0 \times d}$ an input tensor representing an $n$-level tree. For instance, a system of $\ell_2$ equations, padded so that each equation has $\ell_1$ terms which have $\ell_0$ symbols each. In the example above, $\ell_2 = 2$, $\ell_1 = 2$, $\ell_0 = 4$.

We fix a set of embedding dimensions $(d_0, d_1, \ldots, d_{n-1})$ for each of the $n$ levels. For simplicity, one could take $d_i = d \ \forall i$. However, computationally, it makes more sense to consider an increasing

sequence $(d_i)_i$ as we move up the tree since upper levels need to encode more information and require much less compute.

The local self-attention at level 0 (leaf level) is computed as:

$$\mathbf{Y}^{(0)} = \text{Att}(\mathbf{X}^{(0)}\boldsymbol{W}_q^{(0)}, \mathbf{X}^{(0)}\boldsymbol{W}_k^{(0)}, \mathbf{X}^{(0)}\boldsymbol{W}_v^{(0)}) \in \mathbb{R}^{\ell_{n-1} \times \cdots \times \ell_1 \times \ell_0 \times d_0} \tag{2}$$

with $\boldsymbol{W}_{\ell \in \{q,k,v\}}^{(0)} \in \mathbb{R}^{d \times d_0}$ trainable weight matrices.

Subsequently, a pooling operation $p$ is applied to aggregate information and reduce the dimensionality, yielding the input for the next levels:

$$\mathbf{X}^{(i)} = p(\mathbf{Y}^{(i-1)}) \in \mathbb{R}^{\ell_{n-1} \times \cdots \times \ell_i \times d_{i-1}} \quad \forall i > 0 \tag{3}$$

$$\mathbf{Y}^{(i)} = \text{Att}(\mathbf{X}^{(i)}\boldsymbol{W}_q^{(i)}, \mathbf{X}^{(i)}\boldsymbol{W}_k^{(i)}, \mathbf{X}^{(i)}\boldsymbol{W}_v^{(i)}) \in \mathbb{R}^{\ell_{n-1} \times \cdots \times \ell_i \times d_i} \quad \forall i \tag{4}$$

with $\boldsymbol{W}_{\ell \in \{q,k,v\}}^{(i)} \in \mathbb{R}^{d_{i-1} \times d_i}$

In the second phase, information is propagated in a top-down fashion either via cross-attention mechanisms or simply additive aggregations. The latter is trivial and does not impact much the cost analysis, therefore we'll explain the former approach.

At each level, nodes refine their representations by extracting relevant contextual information from their corresponding parent node and their siblings at the upper level:

$$\mathbf{Z}^{(n-1)} = \mathbf{Y}^{(n-1)}$$

$$\mathbf{Z}^{(i)} = \mathbf{Y}^{(i)} + \text{Att}(\mathbf{Y}^{(i)}, \mathbf{Z}^{(i+1)}\boldsymbol{U}_k^{(i)}, \mathbf{Z}^{(i+1)}\boldsymbol{U}_v^{(i)}) \in \mathbb{R}^{\ell_{n-1} \times \cdots \times \ell_i \times d_i} \quad \forall i < n-1 \tag{5}$$

where the queries, keys, and values are viewed as having $\ell_{n-1} \times \cdots \times \ell_{i+2}$ leading dimensions with the remaining $\ell_i \ell_{i+1}$ sequence length for the queries $\mathbf{Y}^{(i)}$ whereas the keys and values (parents) are of sequence length $\ell_{i+1}$. The matrices $\boldsymbol{U}_{\ell \in \{k,v\}}^{(i)} \in \mathbb{R}^{d_{i+1} \times d_i}$ are trainable projection weights.

## 3.3 COST ANALYSIS

To simplify the analysis, we omit the batch size and the notion of multi-head attention, and we neglect some lightweight operations such as pooling in the following calculations. We also use the notation $L_i = \prod_{k=i}^{n-1} \ell_k$ with $L = L_0$ total length of the flattened input.

At level $i$ of the bottom-up phase, the projections $\boldsymbol{W}_{\{q,k,v\}}^{(i)}$ have a complexity of $3L_i \times d_{i-1}d_i$ and the attention is $O(2L_i \ell_i d_i)$. The total complexity per level is:

$$C_{\text{up}}^i = 3L_i d_{i-1} d_i + 2L_i \ell_i d_i$$

For the second phase at level $i$, the projections cost $2L_{i+1} \times d_{i+1}d_i$ and the cross attention call is $O(2L_{i+2} \times (\ell_{i+1}\ell_i) \times d_i \times \ell_{i+1})$. Which adds up to:

$$C_{\text{down}}^i = 2L_{i+1} d_{i+1} d_i + 2L_i \ell_{i+1} d_i$$

Let $\ell_n = 0$ for convenience. The total computational complexity is

$$C = \sum_{i=0}^{n-1} C_{\text{up}}^i + \sum_{i=0}^{n-2} C_{\text{down}}^i \quad \text{with } d_{-1} = d \tag{6}$$

$$= \sum_{i=0}^{n-1} 3L_i d_{i-1} d_i + 2L_i \ell_i d_i + \sum_{i=0}^{n-2} 2L_{i+1} d_{i+1} d_i + 2L_i \ell_{i+1} d_i \tag{7}$$

$$= 3L_0 d_{-1} d_0 + \sum_{i=1}^{n-1} 5L_i d_{i-1} d_i + \sum_{i=0}^{n-1} 2L_i d_i (\ell_i + \ell_{i+1}) \tag{8}$$

By choosing $(d_i)_i$ appropriately, we can control the complexity to be dominated either by the projections or the attention mechanism. It also allows to choose the distribution of the compute over the tree, either allocate most of the compute to the lower level (e.g. $d_i \leq \sqrt{\ell_{i-1}d_{i-1}}$) or to the top level (e.g. $d_i \geq \ell_{i-1}d_{i-1}$) or distribute the compute over the tree $\sqrt{\ell_{i-1}d_{i-1}} \leq d_i \leq \ell_{i-1}d_{i-1}$.

**Case where $d_i = d$:** Complexity is overwhelmingly dominated by the terms $3Ld^2 + 2Ld(\ell_0 + \ell_1)$ while the flat attention's cost is $3Ld^2 + 2L^2d$. When the sequence lengths are larger than the embedding dimension $d$, as when scaling up the inputs, the dominating factors are $(\ell_0 + \ell_1)Ld$ versus $L^2d$. For a regular tree (i.e. an $\ell$-ary tree with $\ell = L^{\frac{1}{n}}$), the cost is $L^{1+\frac{1}{n}}d$ only.

### 3.4 Polynomial Encoding and Tokenization

We consider systems over finite fields $k = \mathbb{F}_q$ (we chose $q = 7$ for all of our experiments) and we adopt the standard tokenization of Kera et al. (2024) without the hybrid embedding. Namely, the vocabulary consists of the union of the sets $\{\texttt{C1}, \ldots, \texttt{Cq-1}\} \cup \{\texttt{E0}, \texttt{E1}, \ldots, \texttt{Ed}\} \cup \{\texttt{<bos>}, \texttt{+}, \texttt{<sep>}\}$ with $d$ maximum degree in the dataset. A polynomial $\sum_u a_u x_1^{u_1} x_2^{u_2} \ldots x_n^{u_n}$ is then tokenized by joining the encodings of each term by the plus token, each term is tokenized by encoding the coefficient and the powers of the variables in each term as $\texttt{Ca}_u \texttt{Eu}_1 \texttt{Eu}_2 \ldots \texttt{Eu}_n$ including the null powers $u_i = 0$.

**Example:** Consider the polynomial system: $p_1 = x_0^2 x_1^2 + 5x_0^2 x_1$, $p_2 = 3x_0^3 + 2x_1$. The tokenization produces the sequence:

```
<bos>  C1 E2 E2  <+>  C5 E2 E1  <sep>  C3 E3 E0  <+>  C2 E1 E0
```

### 3.5 Positional embedding

To encode positional information in multi-dimensional sequential data, we propose a learnable embedding scheme that generalizes standard positional encodings to arbitrary tensor shapes. Given an input of shape $(\ell_{n-1}, \ldots, \ell_0, d)$, we associate each axis $j$ with a dedicated embedding table $E^{(j)} \in \mathbb{R}^{\text{max\_length}_j \times d}$. The positional embedding for a token at index $(i_{n-1}, \ldots, i_0)$ is then constructed as the sum of the corresponding embeddings from each dimension, i.e., $\text{PE}(i_{n-1}, \ldots, i_0) = \sum_{j=0}^{n-1} E_{i_j}^{(j)}$. This approach enables the model to capture hierarchical and multi-axis positional dependencies in a parameter-efficient manner, and can be extended to concatenation followed by a linear projection if desired.

## 4 Experiments

### 4.1 Model Comparison

Figure 2 presents a comparison of different model architectures on the task of predicting Gröbner bases. All models were trained for 52 hours on 8 V100 GPUs using a backward-generated dataset consisting of 1 million multivariate polynomial systems over $\mathbb{F}_7$, with $n = 6$ variables and density $\rho = 0.33$.

The baseline corresponds to the model from Kera et al. (2024), which uses 6 encoder and 6 decoder layers with an embedding dimension of 1024. In contrast, our proposed `HATSolver-2` and `HATSolver-3` models employ the same architectural hyperparameters as the baseline, but replace the standard attention mechanism with our hierarchical attention layer.

In `HATSolver-2`, we use two hierarchy levels. At the lowest level (level 0), the model attends to the individual tokens within a term. At the next level (level 1), it aggregates across all terms within all equations.

`HATSolver-3` introduces a third hierarchy level. Here, level 0 again corresponds to the tokens of a term. Level 1 groups tokens into complete terms within each polynomial, while level 2 aggregates across the set of polynomials.

As shown in the figure, both `HATSolver-2` and `HATSolver-3` demonstrate faster convergence and achieve higher accuracy compared to the baseline throughout training. `HATSolver-2` was faster in training and performed about 450K training steps while the baseline model did less than 300K steps during the trainig period. The case of `HATSolver-3` is particularly noteworthy: although it requires padding—forcing all polynomials to be represented with the same length—the method remains faster than the baseline. The cost of padding is significant here because polynomial lengths vary widely: in this setting ($n = 6, \rho = 0.33$), polynomials range from only 12 terms up to

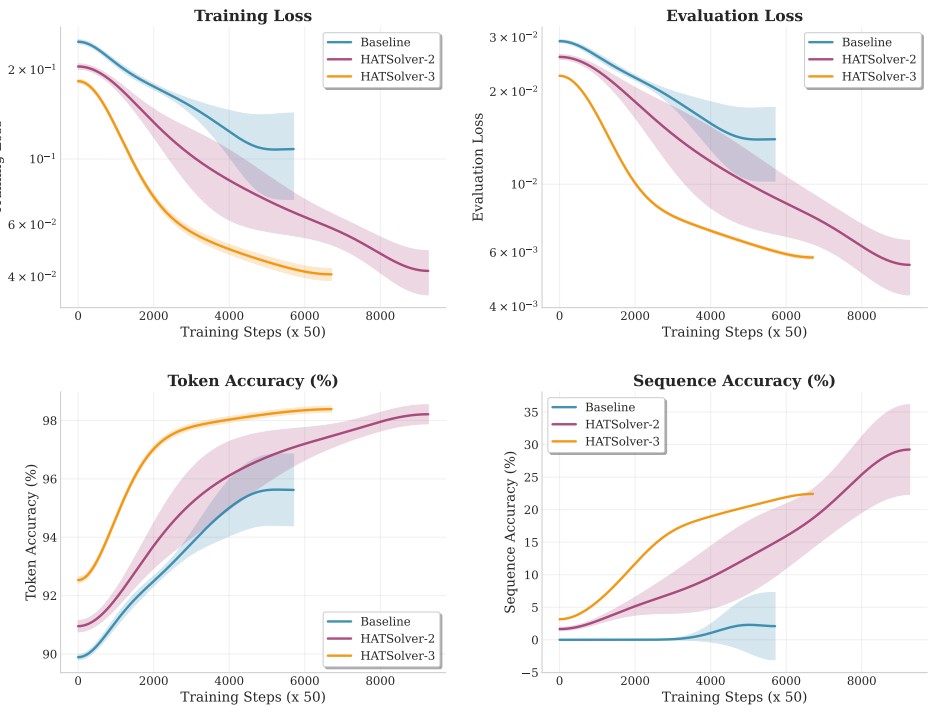

Figure 2: Training dynamics comparison; training models on predicting Gröbner bases for 52 hours on 8 V100 GPUs using the backward generated dataset of size 1 million multivariate systems over $\mathbb{F}_7$ with $n = 6$ variables and density $\rho = 0.33$. Baseline is the Kera et al. (2024) base model with 6/6 encoder decoder layers and 1024 embedding dimensions. The `HATSolver`-2/-3 models are our models with the same hyperparameters as the baseline, with the attention layer replaced by our hierarchical attention layer with 2 and 3 levels respectively. Sequence accuracy refers to the exact match accuracy of the model.

238 terms, with an average of 63 terms. This variability means that shorter sequences are heavily padded, effectively doubling the number of training tokens. Despite this overhead, `HATSolver-3` still maintains a clear advantage over the baseline in both speed and accuracy. More results for $n = 7$ and $n = 10$ including other finite fields such as $\mathbb{F}_{17}$ and $\mathbb{F}_{2^4}$ are reported in the appendix figs. 4 and 5 and appendix C.4.

## 4.2 SCALING UP `HATSolver-3` ($n = 13$)

The instances solved in previous experiments proved to be relatively tractable for classical algorithms such as `STD-FGLM` Greuel et al. (2009); Faugère et al. (1993), as demonstrated in table 11. Hence we now focus on larger and more challenging problem instances. In this section we scale up our model architecture by expanding the embedding dimension to $d = 1408$ while keeping the number of layers to 4/4 encoder/decoder layers. The `HATSolver-3` model is trained on a diverse dataset of 13-variable polynomial systems with various sparsity levels using the curriculum learning schedule discussed in appendix B with $\sigma = 2$ and $v = \frac{1}{2}$. We cap the number of terms per equation in the datasets to 400 to avoid memory spikes. We give some statistics about the datasets in table 7. We use `STD-FGLM` algorithm provided in SageMath with the libSingular backend and `Msolve` as comparison baselines. These algorithms compute the Gröbner basis in the graded reverse lexicographic order (grevlex) which is efficient, and followed by the `FGLM` Faugère et al. (1993) algorithm for the change of term order to the lexicographic (lex) order.

Table 1 presents the performance of `HATSolver-3`, evaluated across a range of systems of equations over $\mathbb{F}_7$. Overall, the results indicate that `HATSolver-3` is capable of learning to compute Gröbner bases and beats the classical algorithms `STD-FGLM` and `Msolve`. It is important to note

| | Density (%) | 30 | 40 | 50 | 60 | 70 | 80 | 90 | 100 |
|---|---|---|---|---|---|---|---|---|---|
| Ours | Success (%) | 52.5 | 49.4 | 47.1 | 49.5 | 55.2 | 56.1 | 61.2 | 60.8 |
| | Support Acc. | 65.0 | 63.0 | 59.0 | 61.0 | 68.0 | 70.0 | 94.0 | 91.0 |
| | Per Token Acc. | 99.8 | 99.7 | 99.8 | 99.8 | 99.8 | 99.8 | 99.9 | 99.8 |
| | Runtime (s) | 224 | 241 | 273 | 277 | 280 | 287 | 292 | 300 |
| STD-FGLM | Success (%) | 33.5 | 19.6 | 10.7 | 8.5 | 8.2 | 7.4 | 6.1 | 6.5 |
| | Runtime (s) | 652 | 936 | 1025 | 995 | 1068 | 687 | 1012 | 1129 |
| Msolve | Success (%) | 32.8 | 17.0 | 11.7 | 6.9 | 7.1 | 4.9 | 4.5 | 4.6 |
| | Runtime (s) | 787 | 773 | 1274 | 766 | 1010 | 852 | 1232 | 588 |

Table 1: Performance of `HATSolver-3` on computing Gröbner bases for polynomial systems with 13 variables over $\mathbb{F}_7$, across varying system densities with comparison to traditional algorithms `STD-FGLM` Greuel et al. (2009); Faugère et al. (1993) and `Msolve` Berthomieu et al. (2021). The density (%) indicates the proportion of nonzero terms in the matrix $U_2$ in backward generation 2.4, controlling the sparsity of the systems. **Success** denotes the exact match test accuracy, i.e. the percentage of the test set instances for which the model generated the exact correct Gröbner basis. **Support Acc.** measures the accuracy when only the support (i.e., the set of monomials) of the polynomials is considered, treating two polynomials as equal if they have the same set of monomials regardless of coefficients. **Per Token Acc.** reports the average token-level accuracy over all outputs. The reported runtime corresponds to the output generation phase, noting that no optimization efforts were implemented (e.g. key-value caching, inference engines, etc.). For the `STD-FGLM` and `Msolve` algorithms, a run is considered failed after a **2 hour** time limit.

that the runtime reported for our model refers to inference time only. Training is performed offline once and does not depend on the number of evaluation samples. The model's exact match accuracy fluctuates with density, exhibiting no simple monotonic trend, but reaches a peak of 61% at 90% density, with support accuracy (see the caption of table 1) reaching 94%. This indicates that challenging instances exist across all levels of sparsity, and that the difficulty of the problem is not solely determined by the density of the system. Notably, when the same training experiment was conducted using the base model, it failed to learn any meaningful solution. The `STD-FGLM` and `Msolve` algorithms on the other hand do not terminate after running for 2 hours for most cases, with up to 93.5% timeout rate for the full density samples and an average runtime of 1129s (resp. 1434s) for the remaining 6.5% (resp 6%) completed runs.

### 4.3 ISOLATING ARCHITECTURAL IMPACT: HATSolver ON $n = 13$ WITHOUT CURRICULUM

To disentangle the contribution of the hierarchical attention architecture from the benefits of curriculum learning, we conducted an ablation study where `HATSolver-3` was trained on a single dataset configuration without any staged progression. Specifically, we trained directly on a dataset of size 1 million samples with $n = 13$ variables over the same finite field at density $\rho = 0.9$.

| Density (%) | 10 | 20 | 30 | 40 | 50 | 60 | 70 | 80 | 90 | 100 |
|---|---|---|---|---|---|---|---|---|---|---|
| Accuracy (%) | 14.40 | 14.31 | 13.45 | 14.72 | 18.36 | 20.11 | 20.10 | 26.42 | **33.85** | 28.57 |

Table 2: (Exact Match) Accuracy of `HATSolver-3` trained *without curriculum* on $n = 13$ variables over $\mathbb{F}_7$ at $\rho_{\text{train}} = 0.9$, evaluated across densities $\rho_{\text{test}} \in \{0.1, 0.2, \ldots, 1.0\}$. Each test set contains 1000 systems. Model architecture: 4 encoder layers, 4 decoder layers, embedding dimension 1408. Bold indicates training density.

**Results and Analysis.** Table 2 reveals three key findings. First, `HATSolver-3` achieves 14–34% accuracy across test densities even without curriculum, demonstrating that the hierarchical architecture alone captures meaningful polynomial structure. Second, performance peaks at the training density $\rho = 0.9$ (33.85%) and degrades with distance from this regime, particularly at low densities ($\rho \leq 0.4$: 13–15%). Meaning that the sparse regime is too far from the training distribution. Third, comparing to curriculum-trained results (Table 1) shows curriculum learning provides a substantial

boost: from 33.85% to 61.2% at $\rho = 0.9$. This confirms that while hierarchical attention provides the representational capacity for $n = 13$ systems, curriculum learning is a complementary technique for better performance. Another factor that could contribute to the performance gap is the amount of training data, as the curriculum-trained model was exposed to more data points than the model in this experiment.

## 4.4 TRAINING ON THE NEW BACKWARD-GENERATED DATA.

We further evaluate `HATSolver` on datasets generated via the backward method of Kera et al. (2025). Note that while this method was designed for Border basis pairs, we utilize it here for Gröbner bases, incidentally validating its use for this task. By sampling random matrices, this approach offers a simpler alternative to the unimodular transformations used previously. It is worth noting, however, that the method is probabilistic: its correctness depends on the field size, which can be a limiting factor on smaller fields.

We train `HATSolver` models with the same architecture on systems with $n = 5$ variables across three different finite fields: $\mathbb{F}_7$, $\mathbb{F}_{16}$, and $\mathbb{F}_{17}$. Results are presented in table 3. The model achieves comparable performance across all three fields, with accuracies ranging from 43-52% on low-density systems ($\rho = 0.2$) and gracefully degrading to 22-33% on full-density systems ($\rho = 1.0$).

Table 3: Exact Match Accuracy (%) of our model on Gröbner basis prediction of multivariate polynomial systems with $\mathbf{n = 5}$ variables across different densities over multiple finite fields. Training is done on 1 A100 GPU per field for 24 hours using the backward data generation method from Kera et al. (2025). Model parameters are 2 encoder layers, 6 decoder layers, 1024 embedding dimensions.

| Field | Density (%) | | | | |
|---|---|---|---|---|---|
| | **20** | **40** | **60** | **80** | **100** |
| $\mathbb{F}_7$ | 52.45 | 48.34 | 44.32 | 37.28 | 33.63 |
| $\mathbb{F}_{16}$ | 43.35 | 36.86 | 35.31 | 25.96 | 24.01 |
| $\mathbb{F}_{17}$ | 44.01 | 43.78 | 32.08 | 29.39 | 22.82 |

## 5 RELATED WORK

Yang et al. (2016) were the first to introduce Hierarchical Attention Networks (HAN) for classifying documents–modeling them as sequences of sentences, which in turn are sequences of tokens. This approach has inspired numerous subsequent works Chalkidis et al. (2019); Wu et al. (2021); Liu et al. (2022); Chalkidis et al. (2022); Yu et al. (2023); Liu et al. (2024); Slagle (2024); Ho et al. (2024). For instance, Wu et al. (2021) introduced Hi-Transformer, a hierarchical interactive architecture for long document processing through a three-stage design. Their approach first employs a sentence Transformer to learn contextual representations for words within each sentence, generating sentence-level representations using special [CLS] tokens appended to each sentence. Subsequently, a document Transformer processes these sentence representations with added positional embeddings to capture global document context and produce document context-aware sentence representations. A third sentence Transformer stage enhances word-level modeling by propagating the global document context back to individual sentences. This is achieved by simply concatenating the global token to the local tokens and applying the transformer on this combined sequence–a process distinct from our second phase. Their complexity analysis demonstrates significant efficiency gains.

The recent work by Videau et al. (2025) introduces AU-Net, an autoregressive U-Net model that integrates tokenization and representation learning into a multi-stage hierarchy operating directly on raw bytes. Unlike traditional fixed tokenization methods such as Byte Pair Encoding (BPE), AU-Net dynamically pools bytes into words and multi-word chunks. This approach eliminates the need for predefined vocabularies and large embedding tables, and allows the model to handle rare or unseen tokens more efficiently.

While these hierarchical Transformer architectures focus on natural language processing tasks, similar hierarchical attention principles have also been adapted to other domains such as computer vision

Liu et al. (2021),Liu et al. (2024), where computational efficiency is critical. Liu et al. (2024) address the computational and memory inefficiencies of standard Multi-Head Self-Attention (MHSA) in vision transformers by introducing the Hierarchical Multi-Head Self-Attention (H-MHSA) mechanism. In their approach, the input image is initially partitioned into small patches, each treated as a token. H-MHSA first computes self-attention locally within small grids of patches to significantly reduce the computational burden. These local features are then merged into larger patches, and global attention is computed over the reduced set of tokens to allow for the modeling of long-range dependencies.

Our model generalizes these prior works, though with distinct structural changes tailored to our problem setting. First, we introduce a top-down cross-attention phase that is absent in standard hierarchical models. For example, where Wu et al. (2021) propagates global context by simply concatenating global tokens to local sequences, we use cross-attention to redistribute context to lower levels. Second, while most existing architectures are restricted to a fixed two-level hierarchy (e.g., local/global or patch/image), our design supports arbitrary tree depths. This is necessary for symbolic computation, where polynomial systems exhibit recursive nesting deeper than typical document or image structures. Finally, we implement this hierarchy within a single, self-contained layer. This allows it to act as a drop-in replacement for standard attention, rather than requiring a specialized network backbone.

Another important direction for introducing inductive bias and improving computational efficiency is through input encoding and tokenization strategies that directly shorten sequence lengths. For instance, Kera et al. (2025) propose a detailed scheme for efficient input sequence representation in polynomial systems. Instead of the standard approach—where each monomial is tokenized into $n + 1$ tokens (one for the coefficient and $n$ for the exponents)—they introduce a "monomial embedding" method, representing each monomial (along with its follow-up token) as a single vector. This reduces the input size by removing the $(n + 1)$ factor present in traditional representations, which is especially significant as the number of variables and degree increases.

## 6 CONCLUSION

In this work, we successfully design and implement a Hierarchical Attention Transformer (HAT) for the task of computing Gröbner bases for multivariate polynomial systems. Our results demonstrate substantial computational improvements over previous results, as well as superior scalability of the `HATSolver` compared to the standard Transformer baseline. Transformer-based solvers may prove to be a good replacement for solving multivariate polynomial equations in the absence of more efficient traditional algorithms. We leave it as future work to investigate whether a model can be trained on any finite field, including non-prime fields such as power-of-two fields (e.g. $\mathbb{F}_{16}$) used in cryptography, and whether the model can generalize to unseen fields. Additionally, some of the techniques explored in this work may offer useful insights for improving HAT models in natural language processing. Further research is needed to assess their practical impact in this domain. A natural extension is to align the training objective with the *computational steps* of the Gröbner basis algorithm, intersecting with the growing field of Neural Algorithmic Reasoning Veličković et al. (2020); Georgiev et al. (2024); Hashemi et al. (2025). Future work should investigate whether `HATSolver` implicitly learns algorithmic primitives—such as S-polynomial construction and reduction—or if explicit supervision on these intermediate steps is required to achieve robust out-of-distribution generalization.

## 7 REPRODUCIBILITY STATEMENT

To ensure reproducibility, we generated all datasets using the publicly available scripts from the repository associated with the prior work we build upon: Kera et al. (2024) [1]. This repository provides detailed instructions and code for dataset creation, enabling other researchers to replicate our data generation process in alignment with previous studies.

We have included a comprehensive list of hyperparameters used in our experiments in Appendix F tables 6 and 12 to facilitate exact replication of our training and evaluation procedures. Our own

---

[1]https://github.com/HiroshiKERA/transformer-groebner

codebase is not publicly available yet. However, we plan to open source it in the future to enable the research community to validate and build upon our results.

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

## A  MATHEMATICAL FOUNDATIONS

### A.1  MONOMIAL ORDERS

Let $k$ be a field and $k[\mathbf{x}] = k[x_1, \ldots, x_n]$ be the polynomial ring in $n$ variables over $k$. A **monomial** in $k[\mathbf{x}]$ is an expression of the form $x_1^{\alpha_1} \cdots x_n^{\alpha_n}$ where $\alpha_i \geq 0$ are non-negative integers. We denote monomials as $\mathbf{x}^\alpha$ where $\alpha = (\alpha_1, \ldots, \alpha_n) \in \mathbb{N}^n$.

A **monomial order** on $k[\mathbf{x}]$ is a total order $\prec$ on the set of monomials satisfying:

1. $1 \prec \mathbf{x}^\alpha$ for all $\alpha \neq 0$
2. If $\mathbf{x}^\alpha \prec \mathbf{x}^\beta$, then $\mathbf{x}^\alpha \mathbf{x}^\gamma \prec \mathbf{x}^\beta \mathbf{x}^\gamma$ for all $\gamma \in \mathbb{N}^n$

The **lexicographic order** $\prec_{\text{lex}}$ with $x_1 > x_2 > \cdots > x_n$ is defined by $\mathbf{x}^\alpha \prec_{\text{lex}} \mathbf{x}^\beta$ if the leftmost nonzero entry of $\beta - \alpha$ is positive. For any nonzero polynomial $f \in k[\mathbf{x}]$, the **leading term** $\text{lt}(f)$ is the largest monomial appearing in $f$ with respect to the chosen monomial order.

### A.2  ZERO-DIMENSIONAL SYSTEMS

Let $I \subset k[\mathbf{x}]$ be an ideal. The **variety** of $I$ is defined as:
$$V(I) = \{\mathbf{a} \in \bar{k}^n \mid f(\mathbf{a}) = 0 \text{ for all } f \in I\}$$
where $\bar{k}$ is the algebraic closure of $k$.

An ideal $I$ is called **zero-dimensional** if its variety $V(I)$ is finite, i.e., $|V(I)| < \infty$. Equivalently, $I$ is zero-dimensional if and only if the quotient ring $k[\mathbf{x}]/I$ is finite-dimensional as a vector space over $k$.

# B   Curriculum Learning Schedulers

We use a structured curriculum learning approach to progressively train our model on systems of increasing complexity. Therefore, we generate datasets $\mathcal{D}_0, \ldots, \mathcal{D}_{n-1}$ of increasing number of variables and increasing densities. During training, we sample from dataset $\mathcal{D}i$ with probability:

$$p(t, i) = \frac{\exp\left(-\frac{(i-\mu(t))^2}{2\sigma^2}\right)}{\sum_{j=0}^{n-1} \exp\left(-\frac{(j-\mu(t))^2}{2\sigma^2}\right)} \quad \text{with} \quad \mu(t) = v \cdot \left\lfloor \frac{t}{\text{steps per epoch}} \right\rfloor$$

where the curriculum center $\mu(t)$ advances with $t$.

The hyperparameters $v$ (learning pace) and $\sigma$ (curriculum width) control the progression speed and overlap between difficulty levels. Setting $v = 1$ advances the curriculum focus by one variable count per epoch, while $\sigma$ determines how much probability mass is distributed to neighboring complexity levels.

Gaussian scheduler ensures smooth and stable transitions between complexity levels while maintaining exposure to simpler problems throughout training. Early training focuses on low-dimensional systems where the model can learn basic reduction patterns, while later stages emphasize higher-dimensional systems that require more sophisticated elimination strategies. The probabilistic sampling prevents abrupt difficulty jumps that could destabilize training.

**Evaluation Protocol**   We evaluate on held-out test sets $\mathcal{T}_0, \ldots, \mathcal{T}_{n-1}$ using uniform sampling across all complexity levels: $p_{\text{eval}}(i) = \frac{1}{n}$ for $i = 0, \ldots, n-1$. This uniform evaluation ensures that performance metrics reflect the model's ability across the full range of problem complexities and not being biased toward the current curriculum focus. At the end of training, we evaluate the model on each dataset separately to characterize the model's performance and scaling behavior.

## B.1   Improving Baseline Model with Curriculum Learning

We trained the base transformer model from Kera et al. (2024) using the curriculum learning approach explained in appendix B over polynomial systems over finite fields. In this experiment, we used a model architecture with 4 encoder and 4 decoder layers and an embedding dimension of $d = 1024$, which differs from the 6-layer encoder/decoder and $d = 512$ configuration used in Kera et al. (2024). This is motivated by our ablation experiments in appendix C.1 which indicate that wider models perform better and train faster than deeper models. The curriculum gradually increased both the number of variables ($n = 2$ to $n = 7$) and the system density ($\rho$), with each training task representing 1 million samples. Unlike the original Kera et al. paper, which only considered up to $n = 5$ and $\rho = 0.2$ and trained one model per configuration, our curriculum enabled the model to learn and generalize to much larger and denser systems at once.

Table 4 reports the model's accuracy in predicting Gröbner bases for multivariate polynomial systems over $\mathbb{F}_7$, evaluated on 1,000 test samples per setting. The curriculum learning approach enables the model to learn beyond the settings considered in prior work, achieving an accuracy of (0.41) for $n = 7$ at full density $\rho = 1$, a significant improvement over the baseline in Kera et al. (2024), which only solved up to $n = 5$ at a much lower density of $\rho = 0.2$.

# C   Complementary experiments

## C.1   The choice of wide versus deep Base Transformer

We present here an ablation experiment on the choice of embedding dimension and number of layers for the base Transformer model. We train wide versus deep models with the following configurations (encoder layers / decoder layers / embedding dimensions) : (3 / 3 / 1024), (4 / 4 / 1024), (6 / 6 / 512), and (8 / 8 / 512) on multivariate datasets with $n = 3$ variables and density $\rho = 0.5$. The training metrics are shown in fig. 3 which indicates that wider models perform better than deeper models for a fixed parameter budget. Based on these results, we choose to use wider base Transformers in our `HATSolver` architecture to maximize learning efficiency.

Table 4: (Exact Match) Accuracy (%) / Support Accuracy (%) of the base transformer model in predicting Gröbner bases for multivariate polynomial systems over $\mathbb{F}_7$. Each cell reports accuracy over 1,000 test samples, after training with 1 million samples per training setting (values shown in red: $(n, *\rho) \in \{(2, 1), (3, 0.5, 1), (4, 0.5, 1), (5, 0.33, 0.67, 1), (6, 0.33, 0.67, 1), (7, 0.25, 0.5, 0.75, 1)\}$. Non-colored cells correspond to interpolated evaluations. $n$ is the number of variables, $\rho$ is the density of the randomly generated system (backward generation method). The curriculum learning approach enables scaling to larger and denser systems compared to prior work.

| | $n$ | | | | | |
|---|---|---|---|---|---|---|
| $\rho$ (%) | 2 | 3 | 4 | 5 | 6 | 7 |
| 10 | 57.1/63.0 | 65.0/71.2 | 53.7/58.7 | 59.0/63.7 | 63.5/69.0 | 50.7/56.3 |
| 25 | 55.6/61.3 | 65.5/72.4 | 61.5/67.2 | 59.9/65.4 | 63.4/69.4 | 46.7/51.2 |
| 33 | 57.0/62.3 | 64.4/71.7 | 59.1/63.6 | 62.3/68.4 | 62.7/67.9 | 50.3/56.1 |
| 50 | 54.0/60.1 | 62.0/69.7 | 57.6/63.8 | 50.5/56.5 | 58.4/62.7 | 43.0/49.3 |
| 67 | 56.0/61.7 | 62.4/69.1 | 51.5/58.3 | 55.3/61.0 | 59.4/66.1 | 38.9/46.7 |
| 75 | 52.4/58.3 | 58.8/66.0 | 55.0/61.1 | 56.9/63.9 | 58.1/66.9 | 42.3/50.9 |
| 100 | 51.3/57.5 | 55.4/63.9 | 53.0/60.6 | 50.4/56.6 | 53.7/61.9 | 42.3/51.3 |

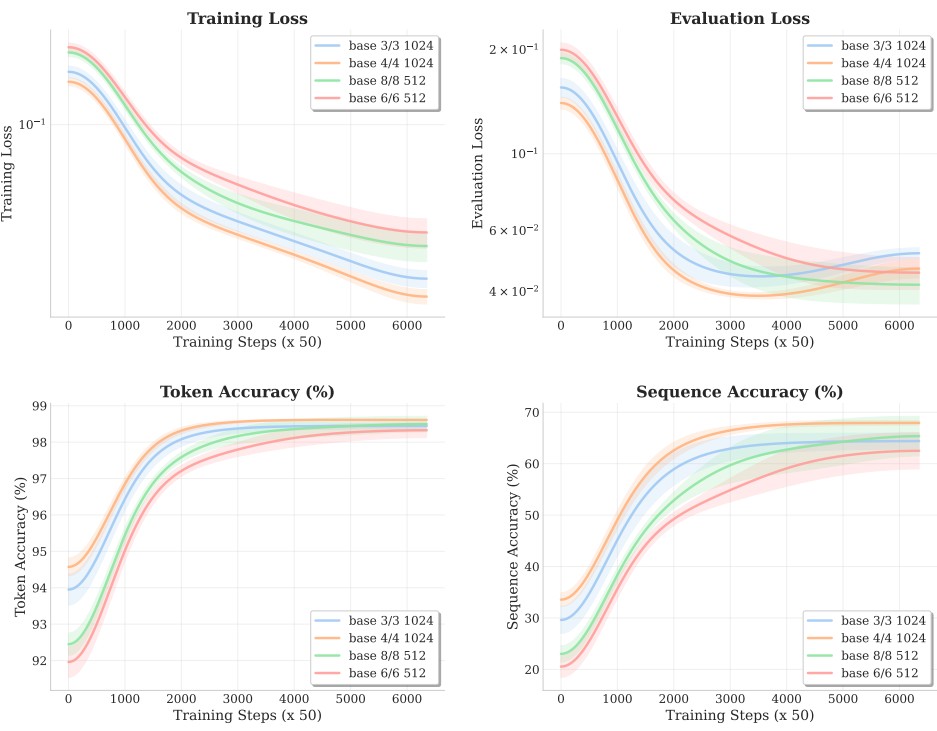

Figure 3: Training base models with varying embedding dimensions and number of layers for 48 hours on multivariate datasets of $n = 3$ variables over finite field $\mathbb{F}_7$ and density $\rho = 0.5$. Each configuration is run with 3 seeds and the plotted lines are smoothened averages. Hyperparameters: batch size = 32, learning rate = $3 \cdot 10^{-5}$ with 1000 warm up steps. Training on $1 \times V100$ per experiment for 48 hours.

## C.2 EXTENDED COMPARISON WITH THE BASE MODEL FROM KERA ET AL. (2024)

We present here more results on the comparison (section 4.1) of `HATSolver` with the base model from Kera et al. (2024). We use the same model size and dataset budget as in section 4.1 but train the models on larger instances, namely the configurations with $n = 7$ variables and density $\rho = 0.25$ and with $n = 10$ variables and $\rho = 0.1$. As a reminder, the model has 6/6 encoder/decoder layers

with $d = 1024$ embedding dimensions. The training was performed on $8 \times$V100 GPUs for 72 hours. The training metrics are shown in figs. 4 and 5 which confirm our findings in section 4.1 that `HATSolver` learns much faster and is computationally more efficient; in fig. 5, the `HATSolver-3` performed more than $400K$ training steps during the 72h training period while the baseline model only did less than $120K$ training steps during the same period. More importantly, the accuracy of the base model is stuck at $0\%$ while `HATSolver-2` and `HATSolver-3` achieve $9\%$ and $8\%$ accuracies respectively.

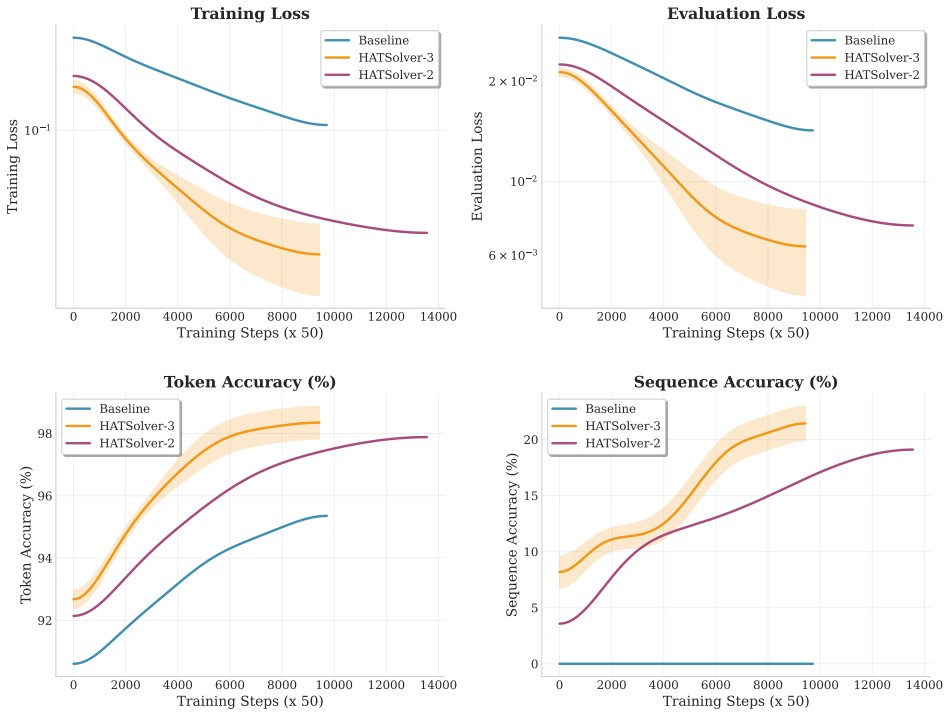

Figure 4: Training dynamics comparison; training models for 72 hours on multivariate datasets of $n = 7$ variables and density $\rho = 0.25$

### C.3    COMPARING WITH HI-TRANSFORMER WU ET AL. (2021)

We implement a version of the Hi-Transformer introduced by Wu et al. (2021) originally designed for long document modeling. Figure 6 shows the architecture of this model. The hierarchy is two levels deep, where a local transformer processes each equation separately, then the first token of each equation is chosen as a representation of the whole equation and passed to the system transformer. The output tokens of the system transformer are put back into their positions within equations. To propagate the global information into local tokens, another series of local transformers is applied to each equation separately.

We ran two experiments over the same datasets from Kera et al. (2024). The first one is over systems of $n = 5$ variables and density $\rho = 0.67$. Another is with $n = 13$ variables and density $\rho = 0.4$ over $\mathbb{F}_7$. The Hi-Transformer show promessing results over the first dataset where it achieved $75\%$ exact match accuracy although at a slower learning pace. For the $n = 13$ case, the model never learned to predict a correct groebner basis. Further more, it's much slower than our 3-level hierarchy `HATSolver-3` both in compute efficiency and learning speed.

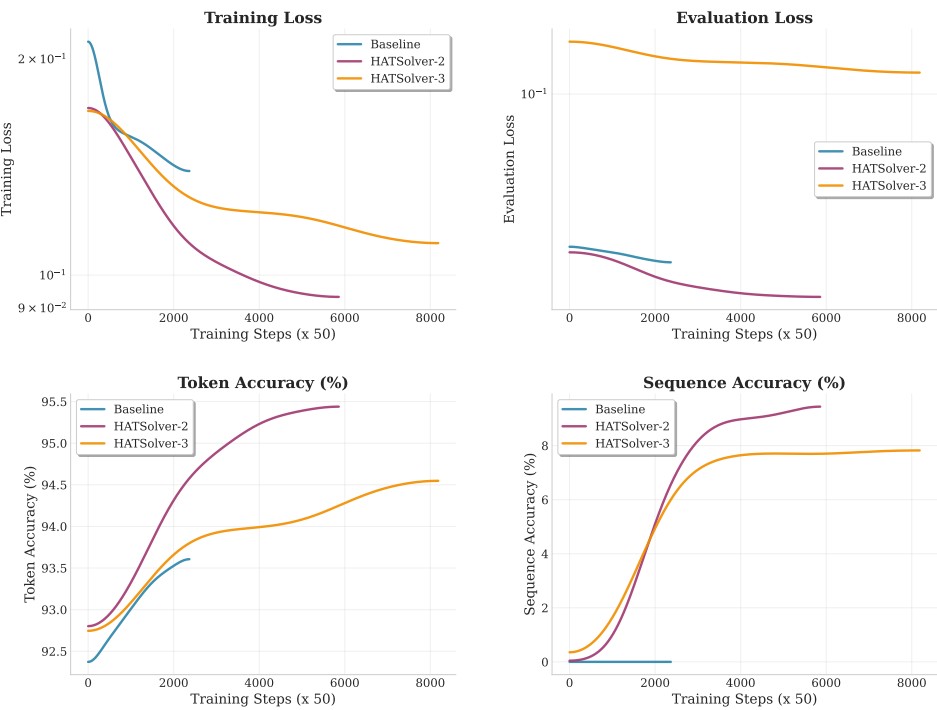

Figure 5: Training dynamics comparison; training models for 72 hours on multivariate datasets of $n = 10$ variables and density $\rho = 0.1$

## C.4 TRAINING HATSolver ON OTHER FINITE FIELDS

While our main experiments focus on the prime field $\mathbb{F}_7$, we investigate the generalizability of HATSolver to other finite fields, including both prime and non-prime fields. We train our model on polynomial systems over $\mathbb{F}_{17}$ (a prime field) and $\mathbb{F}_{2^4} = \mathbb{F}_{16}$ (a non-prime extension field).

For this experiment, we use systems with $n = 7$ variables and evaluate performance across different polynomial densities $\rho \in \{0.2, 0.4, 0.6, 0.8, 1.0\}$. The model architecture consists of 2 encoder layers and 6 decoder layers with embedding dimension $d = 1024$. Training is performed on a single A100 GPU for 24 hours per field, using the same curriculum learning strategy described in appendix B.

Results are presented in table 5. We observe that HATSolver successfully learns to compute Gröbner bases over both prime and non-prime finite fields, achieving comparable performance across both field types. The model reaches approximately 40% exact match accuracy on low-density systems ($\rho = 0.2$) for both fields, demonstrating that the hierarchical attention mechanism generalizes effectively across different field characteristics.

Performance degrades gracefully as density increases, dropping to approximately 27-28% accuracy at full density ($\rho = 1.0$). This trend is consistent across both fields, and probably due to the fact that we stopped training before one epoch using curriculum which assigned most of the sampling probability to the sparsest data bucket ($\rho = 0.2$). The similar performance on $\mathbb{F}_{16}$ (characteristic 2) and $\mathbb{F}_{17}$ (characteristic 17) indicates that HATSolver's learned representations capture fundamental algebraic structure that transcends specific field properties.

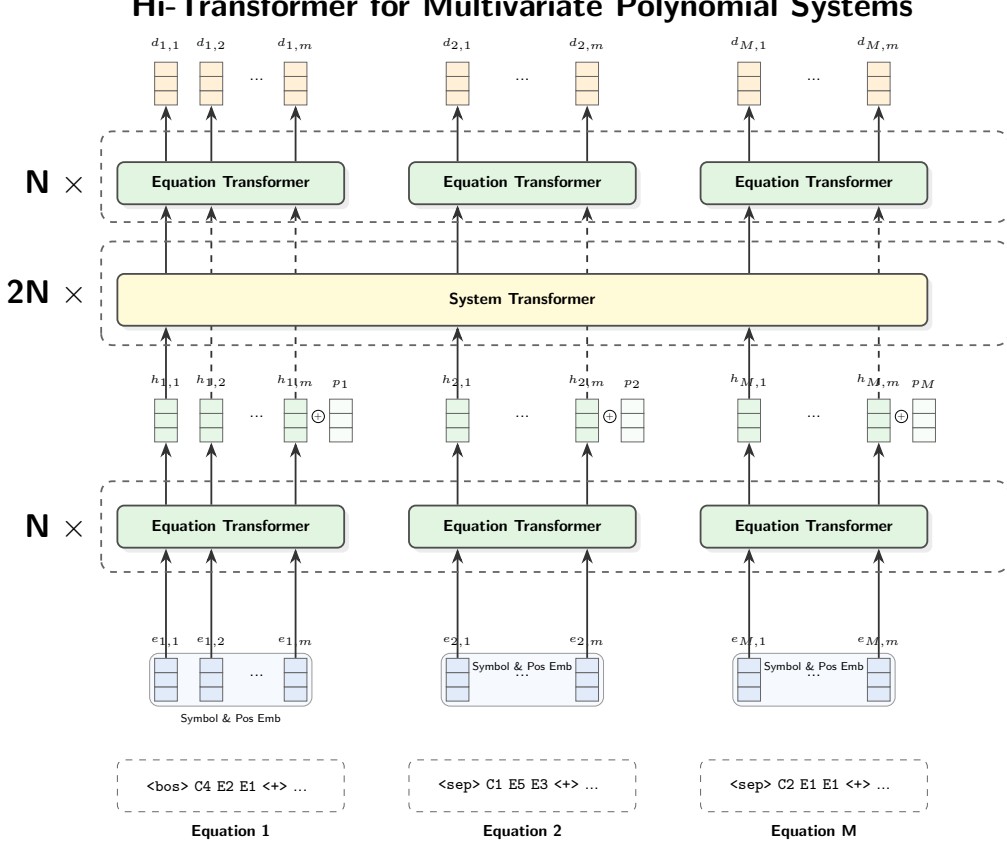

Figure 6: Architectural adaptation of the Hi-Transformer Wu et al. (2021) for multivariate polynomial equation systems. We modified the original layer ordering—originally a repeated sequence of (Sentence, Document, Sentence)—to a grouped configuration of $N$ Equation layers, followed by $2N$ System layers, and finally $N$ Equation layers. This modification ensures that an equal network depth is allocated to both system-wide and equation-level information processing.

Table 5: Exact Match Accuracy (%) of our model on Gröbner basis prediction of multivariate polynomial systems with $\mathbf{n} = \mathbf{7}$ variables across different densities over the prime and non-prime finite fields $\mathbb{F}_{17}$ and $\mathbb{F}_{2^4}$. Training is done on 1 $A100$ GPU per field for 24 hours. Model parameters are 2 encoder layers, 6 decoder layers, 1024 embedding dimensions.

| Field | Density (%) | | | | |
|---|---|---|---|---|---|
| | **20** | **40** | **60** | **80** | **100** |
| $\mathbb{F}_{16}$ | 40.50 | 35.74 | 30.09 | 28.31 | 28.07 |
| $\mathbb{F}_{17}$ | 39.10 | 38.94 | 31.34 | 28.78 | 26.99 |

## D  DATASETS

### D.1  DATA GENERATION CONFIG

Table 6 lists the parameters used to generate the datasets using the script from Kera et al. (2024). We use the same configuration for a fair comparison.

| Hyperparameter | Value | Explanation |
|---|---|---|
| degree_sampling | (empty) | Method for sampling polynomial degrees (not specified) |
| density | $0 < \cdot \leq 1$ | Proportion of nonzero terms in generated polynomials |
| field | GF7 | Finite field used for computations (Galois Field of order 7) |
| gb_type | shape | Type of Gröbner basis generated (e.g., shape position) |
| max_degree_F | 3 | Maximum degree for polynomials in matrices $U_1, U_2$ |
| max_degree_G | 5 | Maximum degree for polynomials in set G |
| max_num_terms_F | 2 | Maximum number of polynomial terms in matrices $U_1, U_2$ |
| max_num_terms_G | 5 | Maximum number of terms in polynomials in set G |
| max_size_F | $n + 2$ | Maximum size (number of polynomials) in set F |
| num_duplicants | 1 | Number of duplicate samples generated |
| num_samples_test | 1000 | Number of samples in the test set |
| num_samples_train | 1,000,000 | Number of samples in the training set |
| num_var | $n$ | Number of variables in the system |
| term_sampling | uniform | Method for sampling the number of polynomial terms (uniform over $[1, \text{max\_num\_terms\_}\{F, G\}]$). |

Table 6: Data Generation Configuration and explanations.

## D.2 STATISTICS OF THE GENERATED DATA

Table 7 provides some statistics about the generated datasets for $n = 13$ variables over finite field $\mathbb{F}_7$. Notably, we report the max_length$_1$ (max sequence length at level 1) which corresponds to the maximum number of terms per equation within a polynomial system (we report the max, mean, and std of this statistics over our dataset) for HATSolver-3. When using HATSolver-2 instead, max_length$_1$ is the total number of terms in a system which we also report.

| Density | Max # Monoms in Sys. | | | Total # Monoms in Sys. | | Max Degree |
|---|---|---|---|---|---|---|
| | max | mean | std | mean | std | max |
| 0.1 | 304.0 | 112.3 | 33.0 | 795.0 | 252.2 | 11.0 |
| 0.2 | 422.0 | 167.6 | 47.8 | 1188.0 | 371.6 | 11.0 |
| 0.3 | 568.0 | 222.7 | 61.6 | 1581.7 | 484.2 | 11.0 |
| 0.4 | 658.0 | 277.4 | 75.0 | 1973.1 | 593.0 | 11.0 |
| 0.5 | 763.0 | 331.9 | 87.7 | 2361.9 | 696.2 | 11.0 |
| 0.6 | 899.0 | 386.3 | 100.0 | 2752.4 | 798.5 | 11.0 |
| 0.7 | 987.0 | 440.9 | 112.6 | 3143.9 | 901.3 | 11.0 |
| 0.8 | 1099.0 | 495.0 | 125.1 | 3531.2 | 1002.9 | 11.0 |
| 0.9 | 1215.0 | 549.0 | 137.4 | 3918.4 | 1104.0 | 11.0 |
| 1.0 | 1269.0 | 602.9 | 149.3 | 4305.4 | 1203.2 | 11.0 |

Table 7: Statistics about the 13-variable datasets over $\mathbb{F}_7$. **Max # Monoms in Sys.** is the maximum number of monomials per equation in each system, for which we report max/mean/std over the whole dataset. This is the metric we cap to 400 for training. This means that for $\rho = 0.4$ for example, we trained on most of the dataset as mean+std $= 277 + 75 < 400$ while for $\rho = 1$, we considered less than 10% of the dataset as mean - std $> 400$. We also report the total number of monomials in each system **Total # Monoms in Sys.** which would be relevant for HATSolver-2 which considers this entity instead.

## D.3 COMPARISON OF BACKWARD GENERATION AND FORWARD GENERATION DATASETS

We represent each polynomial as a vector in a 364-dimensional space, where each dimension corresponds to a monomial up to degree 11. This embedding enables quantitative analysis of the intrinsic complexity and structure of the polynomial systems.

| Metric | Forward Generation | | | Backward Generation | | |
|---|---|---|---|---|---|---|
| | Mean | Std | Median | Mean | Std | Median |
| Sequence Length | 237.51 | 109.62 | 229.00 | 441.69 | 236.62 | 399.00 |
| Max Equation Size | 27.36 | 12.06 | 27.00 | 34.03 | 13.96 | 32.00 |
| Terms per Equation | 15.90 | 12.85 | 14.00 | 22.07 | 14.24 | 19.00 |
| Term Degree | 3.81 | 2.32 | 3.00 | 4.89 | 2.29 | 5.00 |
| Equations per Sample | 3.00 | 0.00 | 3.00 | 4.01 | 0.81 | 4.00 |

Table 8: Statistical Comparison of Forward and Backward Generation Datasets for $n = 3$ over $\mathbb{F}_7$

### D.3.1 MATRIX RANK ANALYSIS

The matrix rank provides insight into the linear independence and intrinsic dimensionality of the polynomial embeddings. Forward generation achieves a rank of 364, while backward generation achieves 216, indicating that forward generation spans the full embedding space while backward generation operates in a more constrained subspace.

### D.3.2 PRINCIPAL COMPONENT ANALYSIS

Table 9 presents the results of our PCA analysis. The number of components required to explain different levels of variance differs between the two methodologies, suggesting distinct underlying structures in the generated polynomial systems.

Table 9: Intrinsic Dimensionality Analysis

| Metric | Forward Generation | Backward Generation |
|---|---|---|
| Matrix Rank | 364 | 216 |
| 90% Variance (dims) | 115 | 104 |
| 95% Variance (dims) | 163 | 128 |
| 99% Variance (dims) | 266 | 173 |
| Sparsity Ratio | 0.956 | 0.939 |
| Avg Non-zero Elements | 15.8 | 22.1 |

### D.3.3 TRAINING RESULTS

Table 10: BART Model Performance on Gröbner Basis Prediction over $\mathbb{F}_7$

| n | Density ($\rho$) | Method | Token Acc. | Acc. | Support Acc. | Samples |
|---|---|---|---|---|---|---|
| 2 | 1.0 | BG | 96.67% | 62.40% | 69.90% | 1000 |
| 3 | 0.5 | BG | 98.50% | 67.38% | 76.19% | 987 |
| 3 | 1.0 | BG | 98.50% | 67.00% | 78.74% | 903 |
| 3 | - | FG | 88.04% | 0.10% | 35.80% | 1000 |

## E CLASSICAL ALGORITHM RUNTIME

To verify the computational complexity claims presented in the Kera et al. (2024), we conducted an independent timing analysis of the `STD-FGLM` algorithm using 1000 trials per configuration for polynomial systems with $n = 2$ to $5$ variables using the same backward generated datasets. Our experimental setup utilized Intel(R) Xeon(R) Gold 6230 CPU @ 2.10 GHz (80 cores) and used the algorithm via SageMath's interface to Singular (ideal.groebner_basis(algorithm='libsingular:stdfglm')). The results reveal substantial discrepancies with the published benchmarks, with our measurements consistently faster than the reported values across all tested configurations. Most notably, for $n = 5, \rho = 0.2$ variables, our mean execution time of $0.008 \pm 0.028$ seconds differs significantly from the claimed $7.46$ seconds. The observed timing variations suggest either different experimental conditions, alternative algorithm implementations or outlier effect on the mean.

| n | Density | Matched | Mean Time ± Std (s) | Median (s) | Max (s) | Paper Claim Mean (s) |
|---|---------|---------|---------------------|------------|---------|----------------------|
| 2 | 1.0 | 100% | 0.005 ± 0.010 | 0.002 | 0.052 | 8.02 |
| 3 | 0.6 | 100% | 0.005 ± 0.010 | 0.002 | 0.050 | 7.50 |
| 4 | 0.3 | 100% | 0.006 ± 0.010 | 0.003 | 0.054 | 7.25 |
| 5 | 0.2 | 100% | 0.008 ± 0.028 | 0.004 | 0.755 | 7.46 |

Table 11: Comparison of `STD-FGLM` algorithm execution times between reported literature values in Kera et al. (2024) and experimental verification. Our results show mean ± standard deviation and median execution times across 1000 trials per configuration (n = 2 to 5 variables, finite field $\mathbb{F}_7$) using 'libsingular:stdfglm' implementation via SageMath. Matched column indicate that the found groebner basis matches the one used in backward generation as a sanity check metric. Our measurements (in seconds) are consistently faster than reported values. All trials were conducted on Intel(R) Xeon(R) Gold 6230 CPU @ 2.10GHz (80 cores).

## F  TRAINING HYPERPARAMETERS

| Hyperparameter | Value |
|----------------|-------|
| Task | multivariate-curriculum |
| Model | hatsolver.3 |
| Eval samples | 1000 |
| Train samples | 1,000,000 |
| Num train epochs | 15 |
| Optimizer | adam_linear_warmup, lr=0.00001, warmup_updates=1000, weight_decay=0 |
| Num encoder heads | 1 |
| Timescale | 40 |
| Clip grad norm | 1.0 |
| Workers | 4 |
| Dtype | float16 |
| Max sequence length | 15,400,15 |
| Max output sequence length | 900 |
| Num variables | 13 |
| Density | -1 # all available densities |
| Train batch size | 2 |
| Val batch size | 2 |
| Field | GF7 |
| Curriculum scheduler ramp | 1 |
| Curriculum scheduler sigma | 4 |
| Curriculum min num variables | 13 |
| Max coefficient | 100 |
| Max degree | 20 |
| Auto find batch size | False |
| Dim expansion per level | 1 |
| Top down cross attn | True |
| Positional encoding combination | concat # versus sum |
| Pad to max length | True |
| Num encoder layers | 4 |
| Num decoder layers | 4 |
| Encoder embedding dim | 1408 |

Table 12: Hyperparameters used in our table 1 training experiment.

