# OpenReview forum: "HATSolver: Learning Gröbner Bases with Hierarchical Attention Transformers"
_ICLR.cc/2026/Conference — ICLR 2026 Oral_

### Official Review · Reviewer_mXdc · 2025-10-30

**Soundness:** 2
**Presentation:** 3
**Contribution:** 2
**Rating:** 4
**Confidence:** 4

**Summary:**

The paper proposes HATSolver, that uses a transformer with hierarchical attention mechanism to capture the hierarchy of he natural tree structure of polynomial systems (tokens  $\rightarrow$ terms $\rightarrow$ polynomials $\rightarrow$ system). The model performs a bottom-up local self-attention followed by a top-down refinement via cross-level attention, and the authors provide a complexity analysis showing how hierarchical factorization reduces the quadratic attention bottleneck as lengths scale. The approach is trained on “backward-generated” datasets that pair non-Gröbner input systems with their reduced lexicographic Gröbner bases in shape position. The paper further adds a curriculum learning schedule to improve scalability. Empirically, the authors show that curriculum learning alone scales the baseline transformer beyond the n≤5 limitation, and demonstrate that their model converge faster and reach higher accuracy than the baseline under equal training budgets.

**Strengths:**

I think adapting hierarchical attention to polynomial systems is a thoughtful inductive bias. Moreover, the results are fairly explorative and include a larger-scale experiment at $n=13$ with a broad density sweep and baselines. On the clarity side, I can say that the two-phase hierarchical mechanism is described clearly with consistent notation. I also believe that scaling and even a partial success in neural Gröbner basis (GB) computation from n≤5 to n=13 is a meaningful step, especially since GB computation is a backbone in many symbolic tasks.

**Weaknesses:**

1. I think the actual success in this task is the exact string equality to the ground-truth reduced lex basis from backward generation. While reduced lex GB is unique, the paper does not report algebraic verification (e.g., reduce every ($f\in F$) by predicted ($\widehat{G}$), check ($\mathrm{lt}(\langle\widehat{G}\rangle)=\mathrm{lt}(I)$), minimality/monicity) as a post-hoc correctness test. The study grades ``exact string equality'', near-misses caused by non-monic scaling, ordering, or tails not fully reduced will be counted as failures, even if the ideal and leading-term ideal are correct. A CAS pass would quantify how many such string failures are algebraically valid solutions. Please add a CAS-based verifier in evaluation.
2. I think the test set is somewhat biased. The scope is restricted to shape position, 0-dimensional radical ideals over ($\mathbb{F}_7$). That matches prior work, but the actual algorithmic alignment requires tests on other fields (incl. non-prime), other monomial orders, and non-shape zero-dimensional ideals. The paper acknowledges this as future work; however, even small OOD tests ($F(5)/F({11}$), random orderings, non-shape samples) would not only strengthen claims, but also would calibrate sensitivity to the data generator and support the beyond ($\mathbb{F}_7$) claim.
3. Since PoSSo and Gröbner basis computation are NP-hard in the worst case, it would strengthen the paper’s soundness and positioning to discuss connections to Neural Algorithmic Reasoning as a neural alignment work (see below). What I really want to see is a comprehensive discussion on what algorithmic primitives (e.g., S-polynomial selection/reduction, Buchberger’s criteria, elimination structure) does your architecture implicitly learn, and how does the hierarchical attention help align with those primitives? A brief discussion would clarify whether HATSolver is merely pattern-matching the training distribution or actually capturing algorithmic invariants that might transfer OOD (e.g., to different fields/orders/densities). Relevant references are:

- Veličković et al. (ICLR 2020), Neural execution of graph algorithms.
- Georgiev et al. (LoG 2023), Neural algorithmic reasoning for combinatorial optimisation.
- Dudzik et al. (LoG 2024), Asynchronous algorithmic alignment with cocycles.
- Hashemi et al. (NeurIPS 2025), Tropical Attention: Neural Algorithmic Reasoning for Combinatorial Algorithms.

**Questions:**

1. Questions discussed in Weaknesses.
2. In Table 2, it’s unclear to me if those numbers are per-instance, per-batch, or total, and for how many instances. Please clarify them.
3. What pooling function you ultimately used (mean? learned? top-k?)
4. Table 4 reports observed max degree 11, while Table 6 lists “Max degree 20” for the training hyperparameters. Clarify the degree cutoffs effective during training vs. evaluation.
5. Please add multiple seeds runs error bars to the reported numbers.
6. Since HATSolver-3 suffers from heavy padding, did you try bucketing by length to mitigate doubled token counts? If so, what was the effect on throughput/accuracy?

---

> ### Author Response · Authors · 2025-11-20
>
> Thank you for your careful reading and thoughtful feedback. Your comments prompted us to take a closer look at several aspects of our work, and we’ve started making changes and running new ablation experiments based on your suggestions. Specifically:
>
> * We will train and evaluate our model on different **finite fields** including non-prime ones such as $\mathbb{F}_8$.
> * We will add a new section to study the **robustness of the model’s outputs to input perturbations** in order to address the question you raised about whether the model is merely pattern matching or actually learning to compute Gröbner bases.
>
> ---
>
> ### Detailed Response
>
> **1. Evaluation Method: Using Computer Algebra System (CAS) Verifier.**
> We appreciate your suggestion regarding the use of a CAS-based verifier. As you noted, Gröbner bases in shape position are particularly straightforward to verify. The support accuracy reported in Table 1 reflects cases where the model correctly identified the monomials but made errors in the coefficients—these are **"near misses"** in the algebraic sense. We are investigating more closely the outputs of the models. We agree that incorporating a CAS-based post-hoc check would provide a more nuanced evaluation and plan to include this in future work where we force the model to output algebraic actions on the system and train it using Reinforcement Learning (RL).
>
> **2. Test Set is Somewhat Biased.**
> We cannot disagree with the referee. The natural follow-up question is whether the test set is actually representative. This issue has been recently investigated in Kambe et al. [1], where the authors introduce a new theoretical framework, namely **Zariski's topology**, to investigate this question. They proved that the generated test set is **dense**, indicating that it is sufficiently diverse. The authors also propose a more general backward generation technique that we are planning to use in further experiments.
>
> **3. Test Set Scope (Generalization to Unseen Fields).**
> We agree that generalizing to unseen base fields, especially larger ones, is non-trivial. To enable the model to handle such cases, additional tokens would be needed to encode field-specific information. We are running an experiment where we **train on different finite fields at the same time** ($\mathbb{F}_5$, $\mathbb{F}_7$, $\mathbb{F}_8$, $\mathbb{F}_{11}$, $\mathbb{F}_{16}$) while adding a field identifier in the input, so the model can distinguish between different fields and field elements (we are using common tokens for the elements of the fields).
>
> **4. Algorithmic Reasoning.**
> Thank you for drawing the connection to **neural algorithmic reasoning (NAR)**. We will add a comparison to the references you provided in our revision. We are also exploring a new direction that makes the model’s actions on the input explicit to better align with algorithmic reasoning.
>
> ---
>
> ### 5. Clarifications
>
> * ***Table 2: Are the numbers per-instance, per-batch, or total? For how many instances?***
>     All reported metrics are computed over total test sets of **1,000 samples per setting**. A test accuracy of 50% means the model predicted 500 solutions exactly right.
> * ***What pooling function was used (mean, learned, top-k)?***
>     In the bottom-up phase of the architecture, we ended up choosing the **first token of each group** to represent the group (those happen to be `<bos>`, `<+>`, and `<sep>`). We experimented with max/mean pooling and didn’t see any performance gain.
> * ***Table 4 vs. Table 6: Why does Table 4 report max degree 11, but Table 6 lists "Max degree 20" for training?***
>     The actual max degree in our data is $\mathbf{11}$. Table 6 is a list of hyperparameters from our training script, and $20$ was chosen arbitrarily high to cover all experiments.
> * ***Did you try bucketing by length to mitigate padding in HATSolver-3? What was the effect?***
>     This is a good catch. We thought about implementing bucketing, which would mitigate the padding issue over batches. However, most of the padding is actually coming from the fact that **within the same system**, the variance of the equation lengths is quite large, which means all equations should be padded to the longest one. We ended up simply filtering out the right tail of the distribution, which partially solves the problem.
>
> ---
>
> **References:**
> * [1] Kambe, Yota Maeda, and Tristan Vaccon. "Geometric generality of transformer-based gr\"{o}bner basis computation." arXiv:2504.12465, 2025.

---

### Official Review · Reviewer_YTxd · 2025-10-31

**Soundness:** 3
**Presentation:** 2
**Contribution:** 4
**Rating:** 6
**Confidence:** 4

**Summary:**

This study presents a new attention-based architecture for computing polynomial systems, particularly Gröbner bases. The prior study, Kera et al. (2024) used a standard architecture and suffered from prohibitively long input sequences and attention memory cost for large polynomial systems. The proposed study introduced a hierarchical attention mechanism that applies attention within terms, polynomials, and systems, accordingly. The experiments show a drastic training cost reduction and extension to systems with a larger number of variables than in the prior study.

**Strengths:**

- A clear focus on scaling up Transformer models for polynomial system computation.
- New hierarchical attention module, which is particularly advantageous for the case where input sequence length surpasses the embedding dimension.
- Experimental results show Transformer models with the proposed attention module scale up the problem size to the size where mathematical algorithms require long computation, which was not the case with the prior work.

**Weaknesses:**

I generally appreciate the achievement of this work - scaling up the problem size of learning-based Gröbner basis computation. The major weaknesses are two-fold: separation from prior studies and clarity/depth of experiments. I suppose that these are manageable weaknesses and expect the rebuttal process will resolve this.

---
**Vs. Prior Studies**

While the proposed concept is clear and aligns with processing polynomial systems, hierarchical attention mechanism has been used in a wide range of applications, as the authors summarize in the Related Work section. It is unclear from the authors' description whether the proposed module is technically novel. For example, MEGABYTE [1], SpaceByte [2], and Block Transformer [3] are general hierarchical attention models. None of them are cited in the paper.

I'd like the authors to present the technical novelty of the proposed method over such works. I understand that the focus of this work is more specialized for polynomial systems. In this context, the proposed method can be regarded as a generalization of monomial embedding [4]. The main text should clarify this as well.

[1] Yu et al., "MEGABYTE: Predicting Million-byte Sequences with Multiscale Transformers," NeurIPS'23

[2] Sagel, "SpaceByte: Towards deleting tokenization from large language modeling," NeurIPS'24

[3] Ho et al., "Block transformer: Global-to-local language modeling for fast inference," arXiv'24

[4] Kera & Pelleriti et al., "Computational Algebra with Attention: Transformer Oracles for Border Basis Algorithms," arXiv'24

---
**Unclear Experimental Setup**

The training setup is somewhat unclear to me.
- In Section 4.1, why was the 4/4 model used rather than the 6/6 one as in the prior study?
- According to the paper, the curriculum learning used 1M samples for each ($n$, $\rho$) combination. The range and step size of $\rho$ are not provided, but the number of training samples seems significant.
- The caption of Table 1 mentions the out-of-distribution evaluations. However, no detailed description can be found. Why do colored texts suggest out-of-distribution? Is that because the corresponding values of $\rho$ were not used in training? Again, the range and step size of $\rho$ are not clarified, but I suspect the "out-of-distribution" $\rho$ here is still interpolation rather than extrapolation.
- I feel the same metric concept is referred to by different names, which made me confused in reading. Specifically, what is the difference between "Accuracy" in Table 1, "Sequence Accuracy" in Figure 2, and "Success" in Table 2?

---

**Experiment Depth**

I suggest a few more experiments that might strengthen the value of the proposed model further.
- [4] proposes a new backward transform method that is more general than the one used in the experiments (adapted from Kera et al.'24). It would also be interesting to evaluate the impact on this setup.
- The visualization of the embedding space of terms, polynomials, and systems would be valuable. For example, are "similar" systems (e.g., having the same leading term set, or Gröbner basis) mapped close together?
- The experiments target polynomial system solving, so the experiments only tested HAT-2 and HAT-3. Setting up an artificial problem/dataset and testing HAT-n with n > 3 could demonstrate the broader utility of this work.

---

I feel the potential practical impact of this work is great, but the machine learning (and algebraic) technical novelty seems limited. My score is temporal, and if the authors can address particularly the first concern (**Vs. Prior Studies**), I'm happy to increase the score.

**Questions:**

Besides the comments in the Weaknesses, I'd like the authors to clarify several points.
- **Dimension design.** [l.263-265] suggests several designs of $(d_i)_i$. Are there any ablation studies and insights into this point?

---

> ### Author Response · Authors · 2025-11-20
>
> Thank you for taking the time to read our paper and share your feedback. Your comments have already helped us improve the manuscript, and they’ve prompted us to run new experiments and clarify several points. Namely:
>
> * We reported an experiment on comparing the **4/4 layers versus 6/6 layers**.
> * We are doing an **ablation on the choice of the hierarchical dimensions** ($d_i$).
> * We will study the validity of **backward generation of Kera et al. 2025 [1]**
>
> Below, we address your questions and suggestions in detail.
>
> ---
>
> ### 1. Technical Novelty vs. Prior Work:
>
> * **Generalization of Hierarchy:** The prior works that we are aware of use a 2-level local-global hierarchy. We sought to provide a more general **multi-level HAT design** that is an easy drop-in replacement of the conventional attention layer. In this sense, our work can be seen as a generalization of previous approaches. The cited works ([1, 2, 3]) modify the overall model architecture, leading to a U-Net-like structure where different layers operate on different sequence lengths.
> * HATSolver achieves modularity by encapsulating hierarchical processing entirely **within a single layer** that serves as a drop-in replacement for standard attention.
> * The **top-down cross-attention** component is a novel element missing in the prior literature.
> * The monomial encoding used in Kera [1] is mentioned in our response to reviewer 1.
>
> We updated our related works accordingly, cited the mentioned references, and highlighted the similarities and differences. We will also add the **block attention baseline** which is available in the literature (e.g., [2] suggestion).
>
> ---
>
> ### 2. Unclear Experimental Setup:
>
> * **4/4 vs. 6/6 Model:** Why was the 4/4 model used in Section 4.1 instead of the 6/6 model from prior work? We found experimentally that **scaling the width is better than scaling the depth**. In practice, we use 4/4 with embedding dimension $d=1024$ instead of 6/6 with $d=512$ because of faster convergence, as shown in Appendix Section C.1 (**The choice of wide versus deep Base Transformer**).
> * **Experiment in Table 1 (Training Data Mix):** The training data includes samples from the $(n, \rho)$ pairs highlighted in red in Table 1. Although each dataset size is $M=1$ million, the curriculum learning assigns a probability $p$ of fetching from that dataset. This probability in practice is less than 0.2 meaning only 20% of the data is trained on.
> * **Interpolation Terminology:** We agree that **"interpolation"** is a more accurate term than "out-of-distribution" in this context. We updated the manuscript accordingly in addition to clarifying the data mix configuration.
> * **Clarifying Metric Definitions:** Thank you for pointing out the confusion regarding "success," "accuracy," and "sequence accuracy." We clarified in our first revision that these terms are used interchangeably for our model.
>
> ---
>
> ### 3. Experimental Depth
>
> * **New Backward Generation from [4]:** We thank the reviewer for bringing this suggestion to our attention. The method from [4] is indeed simple, relying only on sampling a random matrix. We have to verify that it indeed extends to **Gröbner bases**, which are more constrained than Border bases. Although it is surprising that the unimodular condition could be removed, it looks plausible with the additional condition that the generation is correct with some probability. Remark that this probability depends on the field size, which can cause issues for small fields. In any case, the scope of our paper is not on backward generation, and any improved data generation is beneficial for our work. So, we will consider the approach from [4] as well as a new approach, probably more general, from Kambe et al.[3]
> * **Embedding Visualizations:** We acknowledge the value of visualizing the embedding space and are considering this for future experiments hopefully in the next few weeks.
> * **Running HATSolver-4:** While a three-level hierarchy aligns well with the structure of multivariate equation systems, extending to higher levels would require a principled way to group monomials. We are actively exploring approaches to enable this.
> * **Dimension Design Ablation:** [l.263-265] suggests several designs of $(d_i)$. Early experiments did not show any performance gains. As you suggested, we are running more ablation studies, including experiments with $d_0 = 256/128$ and a multiplier of 2. Results will be included in the next revision.
>
> ---
>
> **References:**
> * [1] Kera & Pelleriti et al., "Computational Algebra with Attention: Transformer Oracles for Border Basis Algorithms," arXiv'24
> * [2] Ho et al., "Block transformer: Global-to-local language modeling for fast inference," arXiv'24
> * [3] Kambe, Yota Maeda, and Tristan Vaccon. "Geometric generality of transformer-based gr\"{o}bner basis computation." arXiv 2025.

---

### Official Review · Reviewer_7Nix · 2025-11-02

**Soundness:** 3
**Presentation:** 2
**Contribution:** 2
**Rating:** 4
**Confidence:** 3

**Summary:**

This paper focuses on the recent line of work on learning to compute Gröbner bases with Transformers and proposes to replace flat attention with a hierarchical attention transformer (HAT) that is tailored to the tree shape structure of multivariate polynomial equations. The starting point is the limitation in Kera et al. that the usual Transformers could handle only up to about 5 variables since, in part, the tokenized sequences become extremely long and flat attention is quadratic in sequence length.  The authors propose a HAT layer that is based on hierarchal attention, similar in spirit to U-net based architectures.  The idea is a 2 phase computation where first attention is computed at each layer, pooling to reduce the dimensionality, and in the second phase, information is propagated in a top-down manner via cross addition or residual adds.

**Strengths:**

The motivation compared to Kera et al. is well stated.  The example given to illustrate how their hierarchical model works e.g., in fig 1 is pretty clear and the authors convincingly show that the hierarchical formulation is natural for this data.  They do go beyond the earlier baseline quoted from Kera et al. They claim to reach 13 variables, degree 11, whereas Kera et al. stopped earlier. If those numbers are correct and all other factors are equal, then this is a sizable improvement.

At the same time I should note that I am not a domain expert in learned Gröbner basis computation, so I cannot fully judge how substantial this improvement is from the perspective of the Gröbner basis community, or whether this is mainly a better serialization and positional scheme applied to the same underlying idea.

**Weaknesses:**

The main part of the argument in Section 3 of this paper is that the problem is really the explosion in sequence length and that we should reflect the inherent tree structure in the model. This is reasonable, but it is also very close to what people have been doing with structured tokenization and with position functions that respect the underlying structure in the data, for example with hard ALiBi-like schemes for organized inputs (see eg [1]). The paper does not make it clear why this must be a new hierarchical attention block rather than a careful tokenization and positional encoding strategy that bakes in this inductive bias from the outset.

Second, I am not clear about which part is doing the heavy lifting for the improved performance over Kera et al.  As far as I can tell, there isn't any ablations reported in the paper of whether the performance boost is due to the hierarchical attention, more careful curriculum, different serialization, or training the model longer.

Given how hierarchical the data is, I would have expected at least one baseline that uses grouping of tokens into fixed length blocks with attention only inside blocks plus a cross block layer. Right now the paper is arguing that the custom HAT is needed without showing that a simpler approach fails. This makes the contribution somewhat difficult to judge.  However, I am willing to defer to other reviewers on this point.

Finally, some of the writing is quite informal, particularly in the introduction (such as the first paragraph).  I did not penalize the authors for this.

[1] https://arxiv.org/abs/2402.01032

**Questions:**

(See above)

---

> ### Author Response · Authors · 2025-11-20
>
> Thank you for your thoughtful and constructive review. We are running two additional experiments in response to your feedback:
>
> * To determine the relative impact of the hierarchical architecture vs the effect of the curriculum, we ran a curriculum-less ablation, and this has already been added to the paper (see Section 4.4 and Table 3).
> * We implemented a block attention baseline as you suggested–the experiments are currently in progress and will be included in the next revision of the paper.
>
> More detailed responses to the three weaknesses you identified:
>
> **1. Tokenization versus the new architecture**
> Tokenization is a separate direction for optimizing costs and introducing inductive biases; many papers indeed worked on different tokenization techniques. We chose to optimize the architecture while keeping the input encoding to the finest granularity (one token per symbol) so that no input information is lost. In the case of multivariate polynomials, while changing tokenization might reduce the sequence lengths (e.g. encoding monomials as one token as done in Kera et al.[1]), the impact at scale is limited since the number of monomials grows exponentially with the degree of the polynomials.
>
> **2. Comparing to Hard-ALiBi [2]**
> While approaches like Hard-ALiBi excel at injecting relational bias (e.g. telling the model which tokens are connected in a tree), they do not solve the fundamental bottleneck identified in Kera et al.: Computational Complexity ($O(N^2)$) and Sequence Length. Hard-ALiBi creates hard attention windows to reflect data structure, but (in standard implementations) it still operates on the full $N \times N$ attention matrix. For multivariate polynomials with 13 variables and degree 11, the tokenized sequence $N$ explodes.
>
> **3. Source of Performance Gains**
> We would like to highlight that our baseline comparisons use the same serialization. The "Flat Attention" baseline in our results essentially represents a standard Transformer with these data strategies. The fact that HATSolver outperforms this flat baseline significantly (scaling to 13 variables where the flat model fails to converge for $n=10$ as seen in figure 5 in the appendix) isolates the architecture as the primary driver of scalability. The performance boost is two-fold. First, the improved training speed, which is mainly due to the reduced cost of the HAT architecture. Second, we are able to solve more complex instances mainly due to the architecture. The curriculum helped boost the results to perform across a wide range of settings and obtain better accuracy. To confirm this, we added a section (4.4 and Table 3) about training without using curriculum to isolate the impact of this new architecture.
>
> **4. Block Attention Baseline**
> The Block attention baseline suggested makes sense, the closest configuration to it is the HATSolver.2 without the top-down attention. We are adding an experiment ablation for comparison.
>
> **References:**
> * [1] Kera & Pelleriti et al., "Computational Algebra with Attention: Transformer Oracles for Border Basis Algorithms," arXiv'24
> * [2] Jelassi et al. "Repeat After Me: Transformers are Better than State Space Models at Copying" arXiv'24

---

### Author Response · Authors · 2025-12-02
**Summary of revisions and additional experiments in response to reviewers' questions.**

We thank the reviewers for their helpful feedback.

All three reviewers agreed that the hierarchical attention formulation is natural and well-suited for computing Gröbner bases, a fundamental problem in computer algebra. Reviewer mXdc describes our paper as *"a thoughtful inductive bias"* and Reviewer 7Nix noting that *"the example given to illustrate how their hierarchical model works is pretty clear."* Reviewers also recognized the significance of our experimental results, with Reviewer YTxd highlighting that *"Transformer models with the proposed attention module scale up the problem size to where mathematical algorithms require long computation, which was not the case with prior work,"* and Reviewer 7Nix calling the scaling from $n \leq 5$ to $n = 13$ *"a sizable improvement."*

We uploaded a revised version of the paper (and a diff in supplementary material), in which we addressed all of the reviewers' questions and concerns. In particular, we ran a large number of experiments to show:

- *that our improvements stem from the hierarchical attention mechanism instead of from the curriculum.*
- *that our architecture scales better and outperforms the related work cited by the reviewers.*

---

## Main Concerns Addressed

- **Isolating the main architecture contribution from the curriculum effect:** Since curriculum learning is not our main contribution, we moved the curriculum-related results to the appendix to avoid cluttering the main results section. We also added a results subsection for $n = 13$ without curriculum learning (Section 4.3) to demonstrate that our architecture alone is able to learn at scale.

- **Comparison to prior work:** We expanded the related work section to distinguish our contribution from prior hierarchical attention models and cited additional papers. In particular, we implemented and compared to the Hi-Transformer from Wu et al. 2021[1], which we selected because it is the most applicable architecture to our setting (hierarchical encoder as opposed to hierarchical decoders) and is closest to the block attention transformer idea suggested by the reviewers. The results and a diagram of this architecture are provided in Appendix C.3 comparing with Hi-Transformer (Wu et al. 2021 [1]) and Figure 6. We find that Hi-Transformer performs really well on $n = 5$ variables but as predicted, it doesn't scale to $n = 13$ because the equations themselves become too large. Hence a 3-level hierarchy is necessary in this case of multivariate equations.

- **Applicability to other fields, including non-prime ones:** We conducted experiments on $F_{16}$ and $F_{17}$ to evaluate whether our model can learn from larger and non-prime fields; the results were positive. The model achieved an exact match accuracy of 40% for both the prime field $F_{17}$ and the non-prime field $F_{16}$. This is discussed in Appendix C.4 (Training HATSolver on Other Finite Fields). The next point also addresses this question.

---

## Other Concerns

- **Use of more general backward data generation from Kera et al. 2025 [2]:** We implemented this new and straightforward backward generation method as suggested by reviewer YTxd and generated training datasets ($F_7$, $F_{17}$, $F_{16}$). Ideally, each sampled pair should be verified, as the generation algorithm is only correct with high probability. We assume that the model can tolerate a small portion of corrupted samples in the dataset. We report the results in Section 4.5, where we find that the model is able to learn from this data, although at a slower pace.

- **Wide versus deep transformer ablation:** This question is addressed in Appendix C.1 (The Choice of Wide versus Deep Base Transformer).


[1] Chuhan Wu, Fangzhao Wu, Tao Qi, and Yongfeng Huang. Hi-transformer: Hierarchical interactive transformer for efficient and effective long document modeling. arXiv preprint arXiv:2106.01040, 2021.
[2] Kera & Pelleriti et al., "Computational Algebra with Attention: Transformer Oracles for Border Basis Algorithms," arXiv'24

---

### Meta-Review · Area_Chair_9qEJ · 2025-12-28

**Summary:**

This study introduces a hierarchical attention based architecture for learning to compute Gröbner bases. Building on Kera et al. (2024), which relied on standard transformers and was limited by rapidly growing computational costs, the proposed approach exploits the hierarchical structure of polynomial systems by applying attention at the level of terms, polynomials, and entire systems. Empirically, this design yields substantial cost reductions and enables scaling to problem sizes at which mathematical algorithms require long computation, a regime not reached in the prior work.

Reviewers broadly agree that the inductive bias underlying HATSolver is natural, well motivated, and clearly explained, and that the observed scaling improvements over the Kera et al. baseline are potentially significant. The main concerns focus on clarifying the technical novelty of the hierarchical attention block relative to existing hierarchical or block transformer architectures, and on strengthening the experimental section. In particular, reviewers requested stronger ablations to better isolate and validate the sources of the performance gains, including results without curriculum learning, additional baselines, clearer metrics, and broader evaluation such as experiments over additional finite fields. The authors’ revision and rebuttal substantively address these points. Overall, the paper is assessed as solid and trending positively. One reviewer is already above the acceptance threshold conditional on these clarifications, while the remaining reservations primarily concern the degree of experimental isolation and validation rather than issues of correctness or relevance. Given the nature of the remaining concerns, upward score adjustments are likely, and the revised submission merits acceptance.

**Reviewer Concerns:**

Further experimental requests could be made, including additional ablations or extensions of the newly added experiments, but otherwise no major or acceptance blocking issues appear to remain.

The concerns that have been addressed are indicated in the meta review above.

**Reviewer Scores:**

While exact score changes cannot be known in advance, based on the reviews and the authors’ rebuttal, my assessment is as follows:

Reviewer 7Nix: With the addition of ablation experiments, stronger baselines, and clearer positioning relative to prior work, a modest upward score adjustment above the acceptance threshold is likely.

Reviewer YTxd: Already slightly above threshold and explicitly open to increasing the score; the revisions directly address their main concerns, so an upward adjustment is expected.

Reviewer mXdc: Although initially below threshold, the core concerns focused on experimental depth and validation. The added ablations and broader evaluations make an upward adjustment plausible.

---

### Decision · Program_Chairs · 2026-01-26

Accept (Oral)